# Profiles of oral microbiota and metabolites in periodontitis and benign prostatic hyperplasia patients: a pilot study

Cong Zhu,[1,2,3] Lu-Yao Li,[1,2,4] Cheng Li,[5] Lan Wu,[5] Yuan-Yuan Zhang,[1] Yan Yao,[6] Jun Yang,[1,7] Shuang-Ying Wang,[1] Li-Min Xing,[6] Xian-Tao Zeng,[1,2,3,8] Cheng Fang[1]

**ABSTRACT** Emerging evidence suggests a potential link between periodontitis and benign prostatic hyperplasia (BPH). We have demonstrated that experimental periodontitis and the periodontal pathogen *Porphyromonas gingivalis* exacerbate the progression of BPH. In this study, we focused on the characteristics and alterations of the oral microbiome and metabolites in patients with BPH and those with both BPH and periodontitis. Forty-eight men aged 40 to 60 years were recruited and divided into four groups: healthy controls, periodontitis, BPH, and periodontitis combined with BPH (P-BPH). Gingival crevicular fluid samples were collected for 16S rRNA sequencing and mass spectrometry analysis. The species composition diversity among the four groups showed significant differences. Compared with the healthy group, *P. gingivalis* and *Selenomonas infelix* increased in the P-BPH and BPH groups, respectively. Correlation analysis indicated that *Filifactor alocis*, *Treponema denticola*, and *Porphyromonas endodontalis* were significantly associated with age, International Prostate Symptom Score (IPSS), prostate volume (PV), and periodontal status. Key metabolites, such as arachidonic acid and adrenic acid, were upregulated in BPH and P-BPH groups compared with the healthy group. Our study indicated significant alterations in the oral microbiome and metabolic profile of patients with BPH. Therefore, BPH patients, especially those with periodontitis, may benefit from more active periodontal treatment.

**IMPORTANCE** Benign prostatic hyperplasia (BPH) and chronic periodontitis are age-associated chronic inflammatory diseases that impose a considerable burden on healthcare systems annually. The pathogenesis of BPH remains unclear, with challenges in effective pharmacological management and a high propensity for recurrence. Therefore, clarifying the underlying mechanisms and risk factors for BPH and intervening early in the disease process may potentially delay disease onset and alleviate the associated healthcare burden. Recent research evidence suggests that periodontitis and periodontal pathogens contribute to the development and progression of BPH. The significance of our study is in elucidating the changes in the oral microbiome and metabolites in patients with BPH. By correlating these changes with clinical indicators of patients, we identified key microorganisms and metabolites that may play crucial roles in the process of BPH. This could provide a basis for further mechanistic exploration and disease prevention.

**CLINICAL TRIALS** This study is registered at chictr.org.cn as ChiCTR2300071310.

**KEYWORDS** periodontitis, benign prostatic hyperplasia, microbiome, metabolites

**Peer Reviewers** Jôice Dias Corrêa, Pontifícia Universit Catholic of Minas Gerais, Belo Horizonte, Brazil; Deeksha Tripathi, Central University of Rajasthan, Ajmer, Rajasthan, India

Address correspondence to Li-Min Xing, lmXing1007@163.com, Xian-Tao Zeng, zengxiantao1128@whu.edu.cn, or Cheng Fang, Vitsippa@whu.edu.cn.

The authors declare no conflict of interest.

Periodontitis is a chronic multifactorial inflammatory disease associated with dysbiosis of the plaque biofilm, and it is characterized by the pathological loss of

periodontal ligament and alveolar bone (1, 2). Epidemiological studies have revealed that periodontitis is associated with several systemic diseases (3–5). This relationship may be linked to systemic inflammation, mediated by the hematogenous dissemination of periodontal pathogens or inflammatory mediators, or by their translocation through the body's natural cavities (6, 7). Research on the oral microbiome has identified over 700 bacterial species, in addition to diverse eukaryotes, archaea, and viruses (8, 9). The "red complex," consisting of *Porphyromonas gingivalis*, *Treponema denticola*, and *Tannerella forsythia*, is closely associated with periodontitis (10). Additionally, *Filifactor alocis*, *Peptoanaerobacter stomatis*, and *Saccharibacteria* are also considered potential pathogens (11). Emerging evidence suggests that the oral microbiota is not only associated with oral diseases but also linked to systemic diseases such as cardiovascular diseases, colorectal cancer, and rheumatoid arthritis (12).

Benign prostatic hyperplasia (BPH) is a chronic disease associated with aging and inflammation, with a global prevalence of $1,125.02 \times 10^5$ in 2021 (13). Epidemiological evidence from various regions has revealed a significantly increased risk of BPH in patients with periodontitis (14–16). Periodontal pathogens have been detected in prostate tissue or secretions (17–19). Furthermore, our previous animal experiments have demonstrated that experimental periodontitis and the periodontal pathogen *P. gingivalis* exacerbate the progression of BPH (18). These pieces of evidence highlight the potential role of the oral microbiome in BPH. However, the profiles of oral microbiota and metabolites in BPH patients remain unclear. Therefore, this study utilizes 16S rRNA sequencing and mass spectrometry to compare the oral microbiome and metabolites compositions among BPH, periodontitis, periodontitis combined with BPH, and healthy individuals, aiming to explore potential key hubs in the oral-prostate axis.

## MATERIALS AND METHODS

### Participants and study design

Participants were consecutively enrolled from Xiangyang Integrated Traditional Chinese and Western Medicine Hospital during June to October 2023. This study included men aged 40 to 60 years, who had at least 12 natural teeth and underwent both prostate and periodontal health examinations. Exclusion criteria were a history of prostate surgery, periodontal treatment within the last 3 months, or a history of cancer. The electronic medical records and clinical laboratory test results of the patients were collected for subsequent analysis. The trial was registered at chictr.org.cn (registration no. ChiCTR2300071310).

### Disease diagnosis and sample collection

Considering the patient's medical history, prior treatments, clinical examination results, and the International Prostate Symptom Score (IPSS), professional doctors conducted a diagnosis of BPH. The oral cavity was divided into six segments, with at least one functional tooth selected from each segment, excluding the third molars. If the chosen functional tooth in a segment was missing, it was replaced by a neighboring tooth; if the entire segment lacked functional teeth, it was omitted. Each patient was detected with six segments, a total of six teeth (20). The included teeth were examined for probing pocket depth (PPD), clinical attachment loss (CAL), bleeding on probing (BOP), mobility, calculus, and dental plaque. PPD records the distance from the gingival margin to the bottom of the pocket. CAL is a key indicator to evaluate the severity of periodontitis, determined from the cemento-enamel junction to the deepest site of probing. The diagnosis of periodontitis is based on the latest diagnosis and classification criteria published in 2018 as follows: (i) CAL was detected at the interproximal surfaces of ≥2 non-adjacent teeth, (ii) or buccal or oral CAL ≥ 3 mm, along with PPD ≥3 mm in ≥2 teeth excluding CAL due to non-periodontal causes. Periodontitis is staged based on the severity of the interproximal CAL as follows: Stage I: CAL of 1–2 mm, Stage II: CAL of

3–4 mm, and Stage III/IV: CAL of ≥5 mm (2, 21). All oral parameter assessments were performed by two qualified dental specialists who independently diagnosed and verified the final oral health status of each participant, after a consensus check prior to sample collection to ensure consistency of findings.

After rinsing with sterile saline, the sampling area was isolated with cotton rolls and dried. Gingival crevicular fluid (GCF) samples were immediately collected by the dentist from four sites on six functional teeth, or the alternative teeth with the worse periodontal condition when the specified functional teeth were lost. Specifically, a sterile absorbent paper point was inserted into the buccal, lingual, mesial, central, and distal gingival sulci of the selected teeth and was collected after remaining for 30 s. If the absorbent paper point was blood-stained, it was discarded, and a new sample was taken. The absorbent paper points were placed into 2-mL sterile enzyme-free cryotubes, sealed, and stored at −80°C.

## Bacterial DNA extraction, 16S rRNA gene amplification, and sequencing analysis

DNA was extracted using the MagPure Stool DNA KF Kit B (MAGEN, Guangzhou, China). DNA quality control was performed using a fluorescent dye-based method. This approach quantifies DNA concentration by measuring the fluorescence intensity generated when the dye binds to DNA. The method features simple operation and reliable results, thereby effectively guaranteeing the quality of DNA templates for subsequent PCR amplification. The bacterial rDNA V3V4 variable region was PCR-amplified with primers 338F (ACTCCTACGGGAGGCAGCAG) and 806R (GGACTACHVGGGTWTC-TAAT). During DNA extraction and PCR amplification procedures, we strictly adhered to standardized experimental protocols. Sterile nuclease-free water was systematically included as blank controls to monitor potential contamination throughout the experimental process. PCR products were purified with DNA magnetic beads (BGI, LB00V60). After denaturation, cyclization, and phi29 rolling circle amplification, DNA nano balls were obtained and sequenced on the DNBSEQ-G400 platform (BGI, Shenzhen, China).

Raw sequencing data underwent adapter and primer removal using cutadapt (v2.6), and low-quality reads were filtered using readfq (v1.0), resulting in clean data. Paired-end reads were assembled using FLASH (v1.2.11) to generate hypervariable region tags. These tags were clustered into operational taxonomic units (OTUs) with 97% similarity using USEARCH (v7.0.1090), and chimeras were filtered using UCHIME (v4.2.40) (22). OTU representative sequences were annotated using the RDP classifier (v2.2) with a 0.6 identity threshold. Alpha and beta diversity analyses were performed using the vegan (v2.6.8) and mixOmics (v.6.28.0) R packages. Microbial community functionality was predicted using PICRUSt2 (v2.3.0-b).

We performed Venn diagram analysis of OTUs among the four groups using the VennDiagram (v1.7.3) package in R (v4.4.0). Differential microbial analysis between pairs of groups was conducted using the Wilcoxon test package, with significance set at $P < 0.05$ and $|log_2FC| > 0.32$. Inter-group microbial comparisons were performed using linear discriminant analysis (LDA) effect size (LEfSe) analysis implemented in the microeco package (v1.9.1), with significance thresholds set at LDA score >2 and $P$-value < 0.05. Additionally, the randomForest (v4.7.1.2) package was employed to construct random forest models between pairs of groups to obtain variable importance scores for microorganisms.

## LC-MS/MS analysis

Metabolites were separated and detected using the Waters 2777C UPLC system (Waters, USA) with a BEH C18 column (1.7 μm, 2.1 × 100 mm), alongside the Q Exactive HF high-resolution mass spectrometer (Thermo Fisher Scientific, USA). Mass spectrometry data were imported into Compound Discoverer 3.3 software and analyzed with the BMDB (BGI Metabolome Database), mzCloud, and ChemSpider databases to generate

a data matrix containing metabolite peak areas and identification results (parameters: parent ion mass deviation: <5 ppm, fragment ion mass deviation: <10 ppm, retention time deviation: <0.2 min). Then, the secondary mass spectrum score was used to classify the level and make a credibility assessment of the identified metabolites. Data preprocessing involved normalization (23) and removing compounds with a coefficient of variation (CV) >30% for relative peak areas. Differential metabolite analysis was conducted using the Wilcoxon test in R (v4.4.0) with significance set at $P < 0.05$ and $|\log_2FC| > 0.585$. Heatmaps of differential metabolites were generated using the Pheatmap (v1.0.12) package.

## Statistical analysis

We conducted Spearman rank correlation analysis between metabolites and clinical information, microorganisms and clinical information, as well as microorganisms and metabolites using the corplot package in R (v4.4.0), with a significance threshold set at $P < 0.05$. Various plots and graphs were created using R packages, such as cowplot (v1.1.3), ggstatsplot (v0.12.4), ggplot2 (v3.5.1), ggpubr (v0.6.0), and ggalluvial (v0.12.5). *$P < 0.05$, **$P < 0.01$ were considered significant.

## RESULTS

### Clinical characteristics of the included population

We aimed to explore the differences in oral microbiome and metabolites among patients with periodontitis, BPH, and periodontitis combined with BPH (P-BPH). By adhering to strict diagnostic and inclusion/exclusion criteria, and controlling for risk factors, such as age, smoking, alcohol consumption, and tea consumption, 48 subjects were included in the study. Demographic characteristics and clinical indicators indicate that there were no significant differences among the four groups in terms of age, body mass index (BMI), education level, smoking, alcohol consumption, tea consumption, International index of erectile function (IIEF), prostate specific antigen (PSA) and free prostate specific antigen (FPSA) (Table 1). Compared with the healthy group, the periodontal indexes (BOP, CAL, and PD stage) were elevated in both the periodontitis and P-BPH groups. However, there was no statistical difference in PPD among the four groups, which may be due to the predominance of periodontitis patients with gingival recession. Additionally, the BPH group exhibited significantly higher IPSS, prostate volume (PV), and quality of life (QoL) when compared with the healthy group. The PV and QoL levels in the P-BPH group were also significantly higher than those in the healthy and periodontitis groups (Table S1).

### Periodontitis and BPH affect the composition and diversity of the oral microbiota

A total of 1,506 OTUs were detected, of which 662 were common to the four groups, and 122 OTUs were unique to the P-BPH group (Fig. 1A). Based on the Shannon and Simpson indices, the community diversity in the periodontitis group was significantly higher compared with the Healthy group, BPH group, and P-BPH group (Fig. 1B and C). Principal coordinates analysis (PCoA) and PLS-DA were used to investigate the differences in oral microbial communities among the four groups, which showed that there were statistical differences among the four groups ($P = 0.006$). (Fig. 1D and E).

The species composition of the oral microorganisms for each subject, at the phylum, genus, and species levels, is depicted in Fig. S1A through C. At the phylum level, the four groups were predominantly composed of *Pseudomonadota*, *Bacteroidota*, *Bacillota*, and *Fusobacteriota*, comprising 85.10%–92.38%. Compared with the healthy group, the periodontitis, BPH, and P-BPH groups exhibited a higher proportion of *Bacteroidota* and a lower proportion of *Pseudomonadota* (Fig. S1D). Compared with the Healthy group, the relative abundance of *Acinetobacter* decreased in the periodontitis group and the P-BPH group, while the relative abundance of *Fusobacterium*, *Prevotella*, and *Porphyromonas* increased (Fig. S1E). *Fusobacterium nucleatum* and *Prevotella intermedia* dominated

**TABLE 1** Comparison of clinical parameters among the enrolled population in healthy, BPH, PD, and P-BPH groups[a]

| Parameter | Healthy (*n* = 12) | Periodontitis (*n* = 12) | BPH (*n* = 12) | P-BPH (*n* = 12) | *P* value |
|---|---|---|---|---|---|
| Age | 46.25 (±2.83) | 47.33 (±4.05) | 46.75 (±4.14) | 50.08 (±4.03) | 0.078 |
| BMI | 24.63 (±3.68) | 25.66 (±3.35) | 24.94 (±3.56) | 24.66 (±2.85) | 0.867 |
| Education | | | | | |
| College or above | 6 (50.0) | 6 (50.0) | 6 (50.0) | 3 (25.0) | 0.515 |
| High school or below | 6 (50.0) | 6 (50.0) | 6 (50.0) | 9 (75.0) | |
| Smoking | | | | | |
| Smoker | 5 (41.7) | 3 (25.0) | 2 (16.7) | 4 (33.3) | 0.569 |
| Non-smoker | 7 (58.3) | 9 (75.0) | 10 (83.3) | 8 (66.7) | |
| Drinking | | | | | |
| Non-drinker | 6 (50.0) | 3 (25.0) | 2 (16.7) | 6 (50.0) | 0.200 |
| Drinker | 6 (50.0) | 9 (75.0) | 10 (83.3) | 6 (50.0) | |
| Tea | | | | | |
| Non-tea drinker | 5 (41.7) | 1 (8.3) | 2 (16.7) | 2 (16.7) | 0.208 |
| Tea drinker | 7 (58.3) | 11 (91.7) | 10 (83.3) | 10 (83.3) | |
| PV (mL) | 17.47 (±3.31) | 19.04 (±5.93) | 24.34 (±9.65) | 28.52 (±11.64) | 0.009 |
| IPSS | 2.58 (±2.02) | 3.25 (±2.30) | 9.67 (±5.93) | 7.83 (±9.11) | 0.008 |
| IIEF | 19.25 (±2.96) | 19.42 (±3.26) | 19.00 (±3.44) | 18.82 (±3.89) | 0.975 |
| QOL | 1.42 (±1.44) | 1.92 (±0.90) | 2.83 (±1.27) | 3.08 (±1.44) | 0.008 |
| PSA (ng/mL) | 0.88 (±0.37) | 0.93 (±0.57) | 1.26 (±0.82) | 1.12 (±0.73) | 0.460 |
| FPSA (ng/mL) | 0.23 (±0.09) | 0.25 (±0.20) | 0.30 (±0.18) | 0.30 (±0.17) | 0.614 |
| FPSA/PSA | 0.26 (±0.10) | 0.27 (±0.11) | 0.28 (±0.12) | 0.30 (±0.12) | 0.874 |
| Teeth | 29.00 (±1.60) | 28.75 (±1.71) | 28.75 (±1.86) | 28.58 (±2.11) | 0.956 |
| BOP | 0.12 (±0.17) | 0.35 (±0.26) | 0.09 (±0.10) | 0.53 (±0.29) | <0.001 |
| CAL (mm) | 0.02 (±0.05) | 1.25 (±0.98) | 0.03 (±0.07) | 1.22 (±0.94) | <0.001 |
| PPD (mm) | 2.34 (±0.37) | 2.27 (±0.47) | 2.56 (±0.48) | 2.63 (±0.62) | 0.242 |
| PD stage | 0.00 (0.00) | 2.42 (0.79) | 0.00 (0.00) | 2.00 (0.85) | <0.001 |

[a]Results are mean (±SD) or *n* (%). BMI: body mass index, PV: prostate volume, IPSS: International Prostate Symptom Score, IIEF: International Index of Erectile Function, QoL: quality of life, PSA: prostate-specific antigen, FPSA: free prostate specific antigen, BOP: bleeding on probing, CAL: clinical attachment level, PPD: probing pocket depth, PD stage: periodontitis stage.

the periodontitis and P-BPH groups, whereas *Acinetobacter baumannii*, *Acinetobacter johnsonii*, and *Acinetobacter venetianus* were present in lower proportions compared with the Healthy group (Fig. 1F).

## Comparison of differential microorganisms

The heatmap analysis results indicated that the abundance of *Selenomonas* (*Selenomonas sputigena*, *Selenomonas infelix*), *Capnocytophaga* (*Capnocytophaga granulosa*, *Capnocytophaga leadbetteri*, and *Capnocytophaga ochracea*), and *Neisseria elongata* in the BPH group were significantly elevated compared with the Healthy group. The oral microbiome composition showed significant changes in both the periodontitis and P-BPH groups compared with the Healthy group. Key microorganisms, including *Metaprevotella massiliensis*, *Treponema medium*, *Treponema amylovorum*, *Fretibacterium fastidiosum*, *Treponema denticola*, *Porphyromonas gingivalis*, and *Campylobacter gracilis*, were notably elevated. Additionally, *Tannerella forsythia* and *Filifactor alocis* were significantly increased in the P-BPH group compared to with the Healthy, periodontitis, and BPH groups, while *Vogesella urethralis* and *Bacteroides stercoris* were significantly decreased (Fig. 2). We further conducted LDA effect size analysis to evaluate differences in oral microbiome abundance among the groups. The significantly differential oral microorganism in the BPH and P-BPH groups was *Capnocytophaga granulosa* and *Porphyromonas gingivalis*, respectively (Fig. S2A). Compared with the healthy group, the BPH group exhibited significant differences in abundance of *Selenomonas infelix* and *Capnocytophaga granulosa*, while in the periodontitis group, significant differences were noted in *Porphyromonas*, *Treponema*, *Campylobacter*, *Selenomonas*, and *Capnocytophaga*

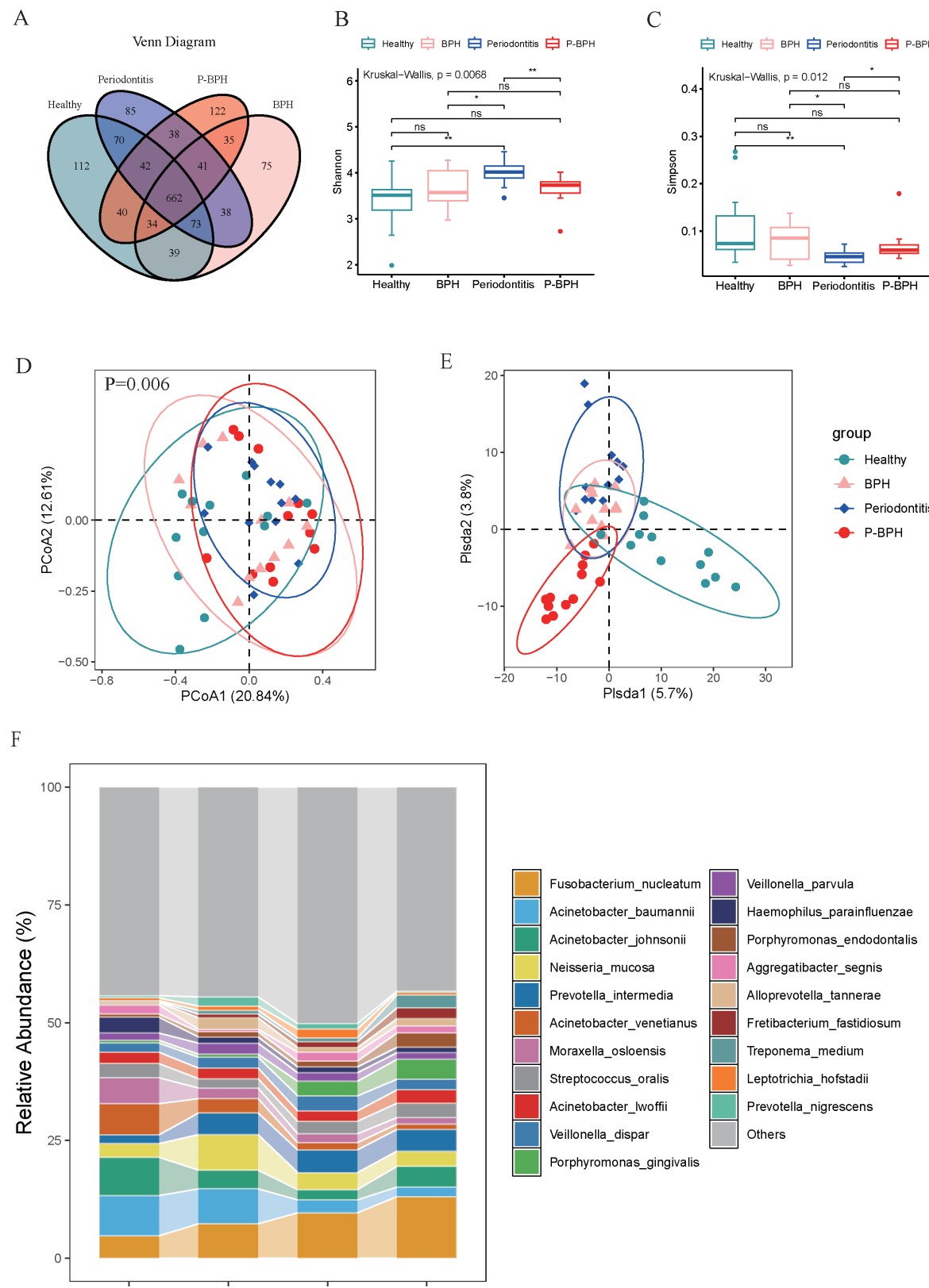

**FIG 1** Comparison of oral microbiota diversity and species composition among the healthy, periodontitis, BPH, and P-BPH groups. (A) Venn diagram of OTUs for the four groups. (B and C) Shannon and Simpson α-diversity of oral microbiota among the groups. (D and E) Bray-Curtis distance-based PCoA and PLS-DA were used to evaluate the β diversity. (F) Species composition at the species level among the four groups. Significant correlations are denoted by * for *P* < 0.05 and ** for *P* < 0.01.

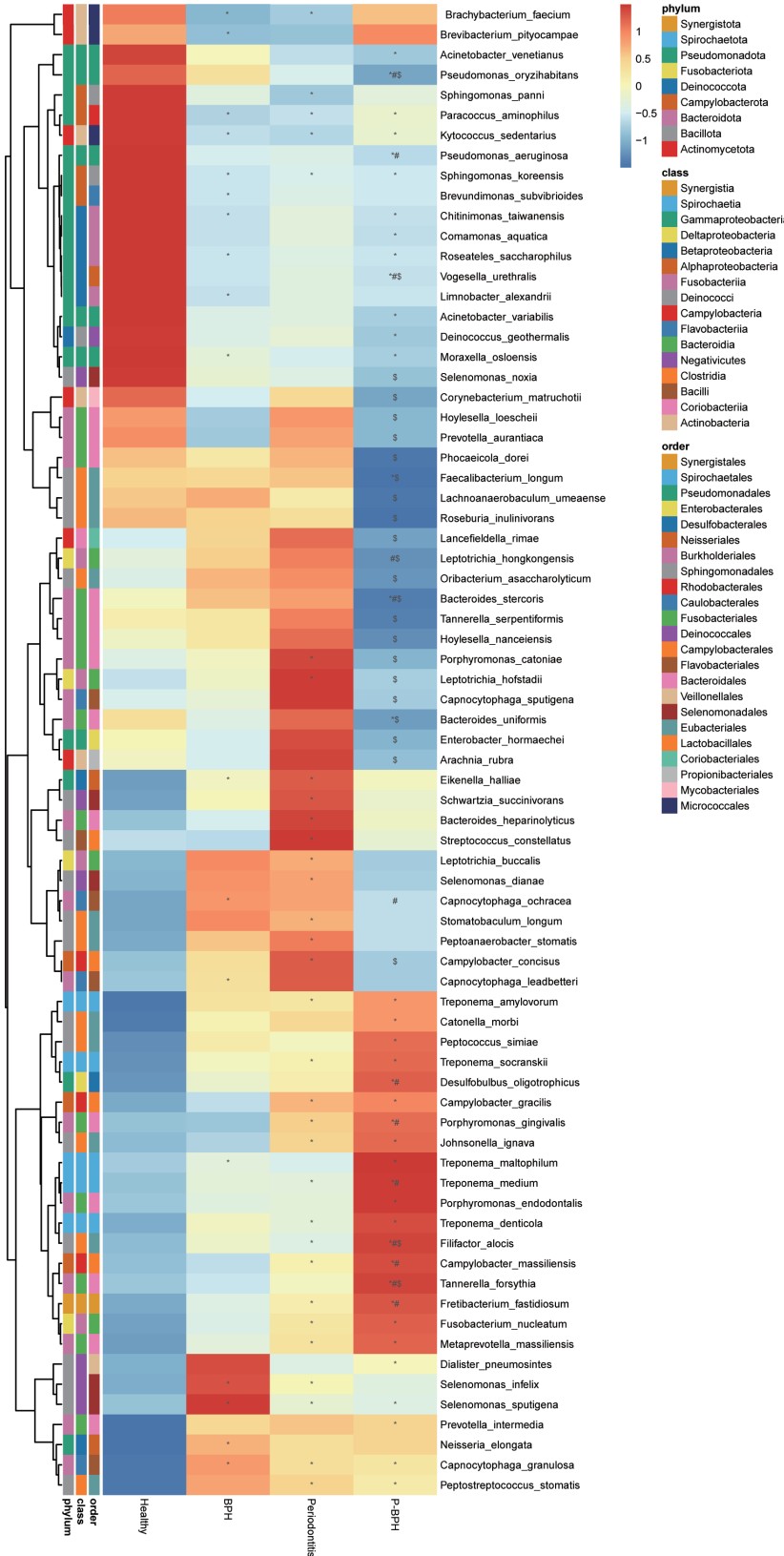

**FIG 2** Heatmap of the significantly differential microorganisms among the four groups. *, compare with the Healthy group; #, compare with the BPH group;, and $ compare with the periodontitis group.

(Fig. S2B and C). When comparing the P-BPH group to the BPH group, *Porphyromonas gingivalis* and *Pseudomonas oryzihabitans* were the significantly differential microorganisms (Fig. S2D).

## Association between differential microorganisms and clinical indicators

We conducted Spearman correlation analysis between the relative abundances of microbial species that were significantly different between any two groups above and clinical indicators. Notably, we identified 22 microorganisms associated with IPSS, half of which are also linked to QoL. Microorganisms significantly positively correlated with PV included *Fretibacterium fastidiosum*, *Desulfobulbus oligotrophicus*, *Tannerella forsythia*, *Dialister pneumosintes*, and *Johnsonella ignava*. Among these, *Fretibacterium fastidiosum*, *Desulfobulbus oligotrophicus*, and *Tannerella forsythia* were also positively correlated with QoL, and *Dialister pneumosintes* showed a positive correlation with IPSS. (Fig. 3A). In addition, we observed that *Arachnia rubra* was uniquely correlated with both IIEF and PSA. Specifically, it demonstrated a positive correlation with IIEF and a negative correlation with PSA.

Among all clinical indices, the proportion of differential microorganisms significantly associated with periodontitis stage was the highest at 40.54%, followed by IPSS (29.73%), CAL (27.03%), and age (25.68%). Microorganisms significantly positively associated with age, periodontitis, and BPH-related indicators include *Fretibacterium fastidiosum*, *Desulfobulbus oligotrophicus*, *Filifactor alocis*, *Schwartzia succinivorans*, *Treponema denticola*, *Prevotella intermedia*, *Treponema medium*, *Tannerella forsythia*, and *Porphyromonas endodontalis*. Among the nine aforementioned microorganisms, four displayed significant negative correlations with high-density lipoprotein. Notably, *Fretibacterium fastidiosum* was also significantly positively correlated with triglycerides (Fig. 3A).

To further identify the key microorganisms in distinguishing diseases, we employed random forest analysis. The results revealed that the primary key microorganism in the BPH and periodontitis group compared with the Healthy group was *Capnocytophaga granulosa* and *Sphingomonas panni*, respectively (Fig. 3B and C). Additionally, the key microorganism distinguishing the P-BPH group from the BPH and periodontitis groups was *Tannerella forsythia* and *Fusobacterium periodonticum*, respectively.

## Metabolites had different compositions among the four groups and were related to clinical indicators

Next, we delved into the metabolite characteristics of GCF across these groups. A total of 547 metabolites (8%) were ultimately excluded. These excluded high CV features primarily represent low-abundance or noise-prone compounds with unstable quantification. In contrast, retaining CV ≤ 30% features significantly improved statistical power and reproducibility, ensuring reliable differential metabolite identification and pathway analysis. Compared with the Healthy group, we identified 36, 14, and 73 differential metabolites in the BPH, periodontitis, and P-BPH groups, respectively, spanning 23 distinct categories. In the BPH group, the significantly upregulated metabolite categories included amino acids (including valylleucine, lysylleucine, and histidyltyrosine), fatty acyls (including mevalonic acid, adrenic acid, and 8Z,11Z,14Z-eicosatrienoic acid) and pyridine and derivatives (including nicotinic acid, orotic acid, and thiamine). The key differential metabolites in the P-BPH group compared with the BPH group included oleandomycin, uric acid, and MG (0:0/16:1(9Z)/0:0). Compared with the periodontitis group, the main differential metabolites were 5-acetylamino-6-amino-3-methyluracil, nervonic acid, and o-phosphoethanolamine (Fig. 4A; Table S3). Among these differential metabolites, 54 (56.25%) were associated with QoL, 47 (48.97%) with age, and 43 (44.79%) with BOP. Additionally, 22 metabolites showed significant positive correlations with IPSS, including valylglycine, orotic acid, N1, N8-diacetylspermidine, and nicotinic acid, which were also significantly positively correlated with age and QoL (Fig. S3A).

Compared with the Healthy group, the P-BPH group exhibited 73 different metabolites, with 58 upregulated and 15 downregulated. The largest categories of these

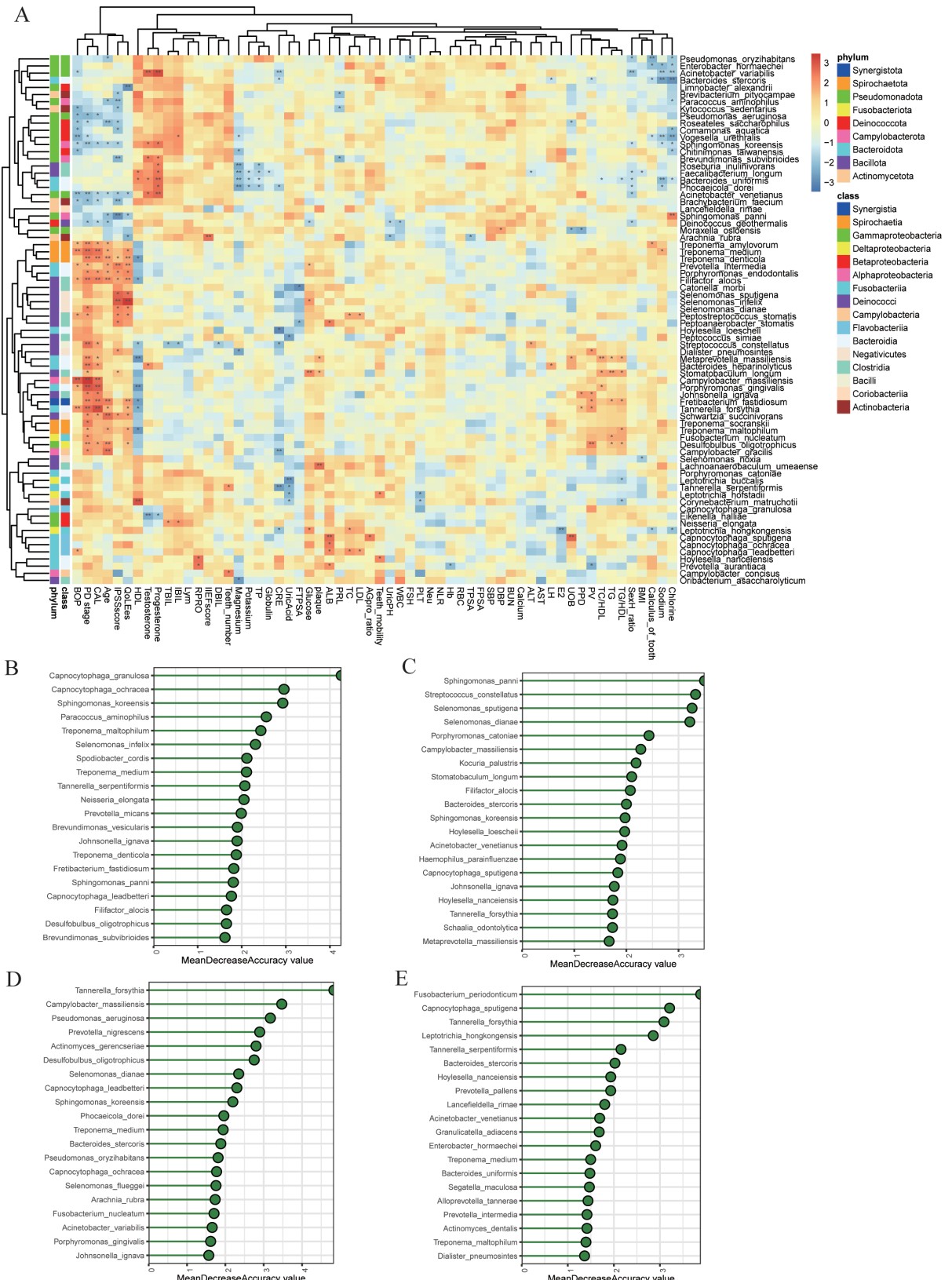

**FIG 3** Correlation between differential microorganisms and clinical parameters. (A) Heatmap of Spearman correlation coefficient between four groups of oral differential microorganisms and clinical information. A random forest model was used to identify the most important microbial features in various comparisons: (B) BPH versus Healthy; (C) periodontitis versus Healthy; (D) P-BPH versus BPH; (E) P-BPH versus periodontitis. BMI: body mass index, SexH_ratio: estradiol to (Continued on next page)

Fig 3 (Continued)

testosterone ratio, TG: triglycerides, HDL: high-density lipoprotein, TC: total cholesterol, PV: prostate volume, PPD: probing pocket depth, UOB: urine occult blood, E2: estradiol, LH: luteinizing hormone, AST: aspartate transaminase, ALT: alanine transaminase, BUN: blood urea nitrogen, DBP: diastolic blood pressure, SBP: systolic blood pressure, FPSA: free prostate-specific antigen, TPSA: total prostate-specific antigen, RBC: red blood cell count, NLR: neutrophil to lymphocyte ratio, Neu: neutrophil percentage, PLT: platelet count, FSH: follicle-stimulating hormone, WBC: white blood cell count, UricPH: urine pH, AGpro_ratio: albumin/globulin, LDL: low-density lipoprotein, PRL: prolactin, ALB: albumin, FTPSA_ratio: FPSA/TPSA, CRE: creatinine, TP: total protein, DBIL: direct bilirubin, IIEF: international index of erectile function, RPRO: urine protein, Lym: lymphocyte percentage, IBIL: indirect bilirubin, TBIL: total bilirubin, QoL: quality of life, IPSS: international prostate symptom score, CAL: clinical attachment loss, PD stage: periodontitis stage, BOP: bleeding on probing.

metabolites were amino acids, peptides, and analogs, as well as fatty acyls, each representing 28.77%. Significantly upregulated metabolites in the P-BPH group included adrenic acid, threonylphenylalanine, leucylleucine, oleandomycin, histidyltyrosine, and phenylalanylphenylalanine. Additionally, uric acid levels in the P-BPH group were markedly higher than in the other three groups (Fig. 4A; Table S2). The random forest analysis results indicated that the key metabolites distinguishing the periodontitis, BPH, and P-BPH groups from the healthy group were MG [18:1(9Z)/0:0/0:0], abietic acid, and hexamethylenetetramine, respectively (Fig. S3B through D). To distinguish from the P-BPH group, the key metabolites for the BPH group and the periodontitis group were 4-methylcatechol and stearic acid, respectively (Fig. 4B and C).

## Association of differential microorganisms with metabolites and microbial function prediction

Further association analysis was conducted to explore the relationship between differential metabolites and differential oral microorganisms. Among the 74 differential microorganisms, *Filifactor alocis*, *Peptostreptococcus stomatis*, *Fretibacterium fastidiosum*, *Porphyromonas endodontalis*, *Treponema medium*, *Treponema denticola*, *Metaprevotella massiliensis*, and *Selenomonas infelix* showed significant positive correlations with 30% to 60% of the metabolites. These metabolites were primarily amino acids, peptides, and analogs, including N-acetyl-L-aspartic acid, valylglycine, N6-acetyl-L-lysine, and D-alanyl-D-alanine, as well as N1, N8-diacetylspermidine, thiamine, hydroxyphenyllactic acid, and N-acetyl-D-glucosamine. On the other hand, methylglutaric acid exhibited a significant negative correlation with these eight microorganisms. Correspondingly, *Roseateles saccharophilus*, *Paracoccus aminophilus*, *Kytococcus sedentarius*, *Chitinimonas taiwanensis*, and *Comamonas aquatica* showed significant negative correlations with 41% to 75% of the metabolites. In contrast, these microorganisms were significantly positively correlated with methylglutaric acid, adipic acid, ethyl 3-hydroxybutyrate, trans-anethole, valeric acid, mevalonic acid, 3-hydroxydecanoic acid, 12-hydroxydodecanoic acid, and genipin (Fig. 5A).

There were 192 KEGG function pathways predicted by PICRUST2, 27 of which were significantly different among the four groups. These pathways primarily involve amino acid metabolism (C5-branched dibasic acid metabolism, cysteine and methionine metabolism, valine, leucine, and isoleucine biosynthesis, phenylalanine, tyrosine, and tryptophan biosynthesis), lipid metabolism (glycosphingolipid biosynthesis, glycosyl-phosphatidylinositol (GPI) anchor biosynthesis, ubiquinone and other terpenoid-quinone biosynthesis, glyoxylate and dicarboxylate metabolism), energy metabolism (glycolysis/gluconeogenesis, butanoate metabolism, pentose and glucuronate interconversions), hormone and vitamin metabolism (ascorbate and aldarate metabolism, steroid hormone biosynthesis), nucleic acid metabolism and repair (non-homologous end-joining), material transport and absorption (ABC transporters, protein digestion and absorption, vasopressin-regulated water reabsorption), signal transduction and regulation (apoptosis, inositol phosphate metabolism), and other metabolic pathways (bacterial invasion of epithelial cells, tropane, piperidine, and pyridine alkaloid biosynthesis, nitrogen metabolism, etc.) (Fig. 5B).

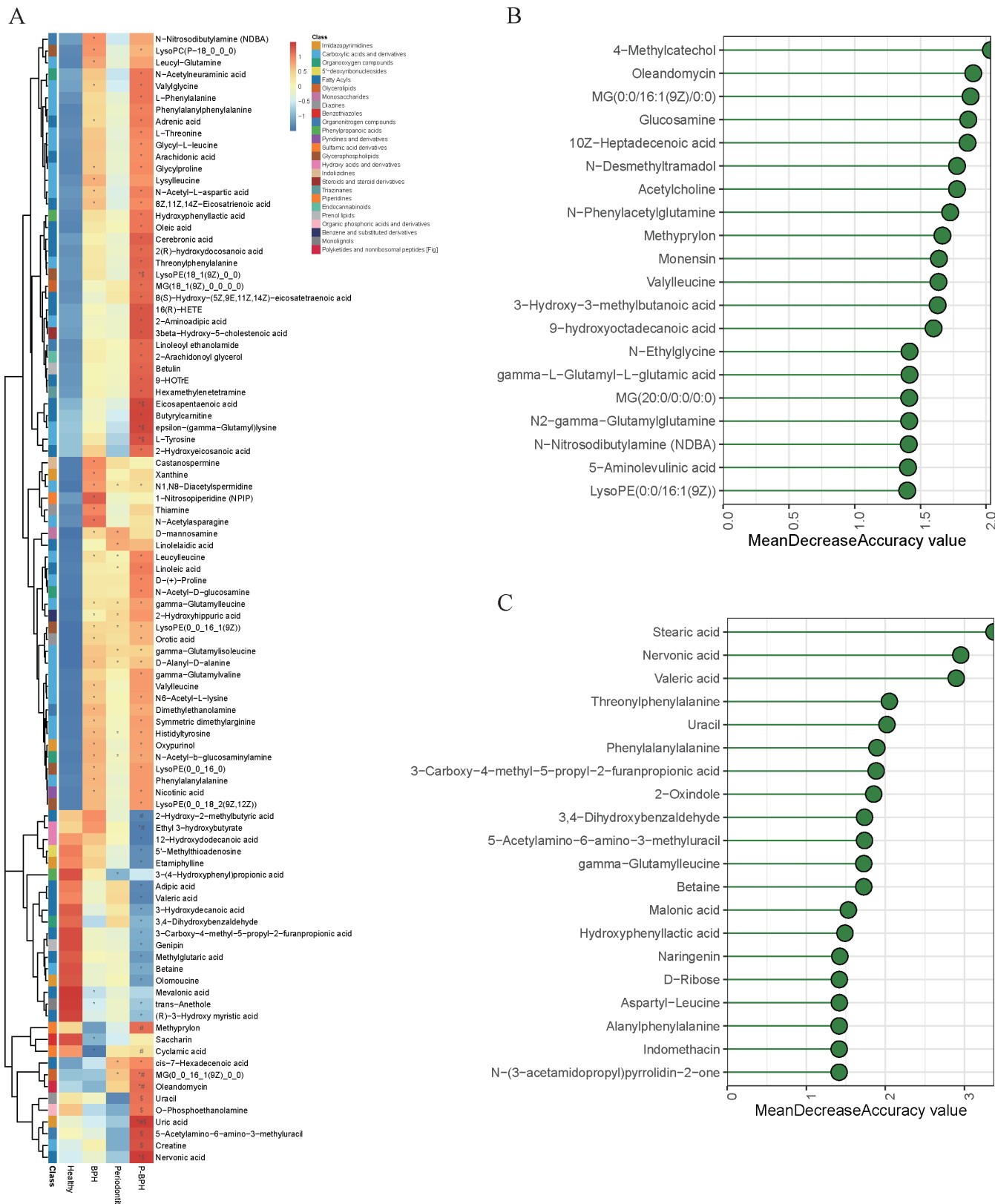

**FIG 4** Gingival crevicular fluid metabolite distribution and differences between the health, periodontitis, BPH, and P-BPH groups. (A) The heatmap of metabolite distribution differences among the four groups. A random forest model was used to identify the most important metabolic features in various comparisons: (B) P-BPH versus BPH; (C) P-BPH versus periodontitis.

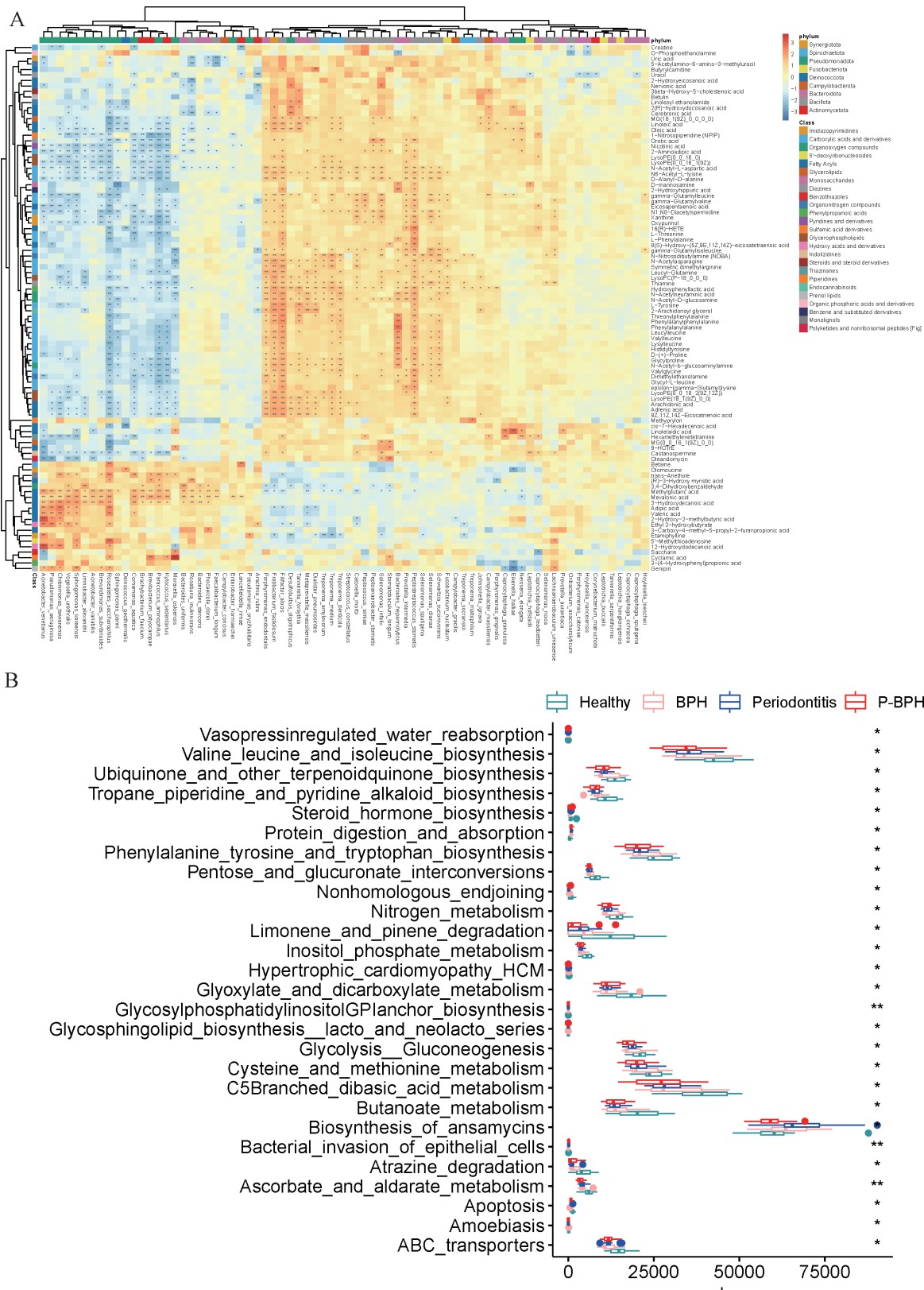

**FIG 5**  Differential microorganisms and metabolites association analysis and microbial function prediction. (A) Correlation analysis between differential metabolites and differential microorganisms. (B) Differential analysis of KEGG pathways for microbial function prediction among the four groups.

## DISCUSSION

In previous studies, investigators detected periodontal pathogens such as *Porphyromonas gingivalis*, *Treponema denticola*, and *Capnocytophaga ochracea* in both the dental plaque and prostatic fluid of patients concurrently suffering from BPH and periodontitis (17, 18). However, methodological constraints in detection approaches and sample stratification prevented comprehensive comparisons of oral microbiome and metabolite profiles across patient subgroups, those with comorbidities, single disease presentations, and healthy controls.

Our investigation into the alterations in oral microbiota and metabolites in periodontitis patients, with or without BPH, revealed significant changes in those suffering from both diseases compared with healthy individuals or those with only one condition. We identified oral microorganisms and metabolites significantly correlated with IPSS and PV, suggesting that changes in oral microbiota and metabolites may be a crucial link between periodontitis and BPH. Pathways related to oxidative phosphorylation and amino acid synthesis and metabolism might play a role in the "oral-prostate axis" (18).

Our findings show a decrease in *Pseudomonadota* and an increase in *Bacteroidota* in the oral microbiota of patients with periodontitis, BPH, or both, compared with healthy individuals. This trend aligns with observations from other studies (24–26). We observed a significant increase in *Porphyromonas gingivalis* and *Treponema medium* in the P-BPH group. Our published findings demonstrated that *Porphyromonas gingivalis* infection significantly upregulates IL-6 expression in prostate tissue and activates the PI3K/AKT signaling pathway, thereby modulating prostate cell proliferation and apoptosis (18). *Treponema medium*, a Gram-negative anaerobic spirochete, has been detected in dental plaque from periodontitis patients. However, direct evidence linking this oral pathogen to BPH pathogenesis remains lacking (27). In this study, *Fusobacterium nucleatum* and *Prevotella intermedia* were the most prevalent pathogens in the periodontitis and P-BPH groups, with notably higher proportions compared with the Healthy group. *Prevotella intermedia* is widely recognized as a major periodontal pathogen and is frequently detected in subgingival samples of periodontitis patients (28). *Prevotella intermedia* plays a crucial role in the onset and progression of periodontitis by inducing various proinflammatory cytokines, proteases, and matrix metalloproteinases (MMPs) (29–31), and is positively correlated with the severity of periodontitis (32, 33). *Fusobacterium nucleatum* is an opportunistic pathogen commonly found in the oral cavity (34). It is associated with various periodontal diseases (35), has the capacity to colonize other tissues and organs through multiple pathways (36), and is one of the most prevalent extraoral bacteria in diseased states (37). An increasing number of studies have discovered that these two oral microorganisms are linked to diseases in other systems, including adverse birth outcomes (38), Alzheimer's disease (39, 40), bacteremic pneumococcal pneumonia (41, 42), atherosclerosis (43), and colorectal cancer (44). We also observed a significant positive correlation between *Prevotella intermedia* and IPSS, suggesting a potential link between this microorganism and BPH.

In patients with prostatic hyperplasia, we observed a significant increase in *Selenomonas* and *Capnocytophaga* in their oral cavity compared with the normal population. *Selenomonas*, a genus within the family *Selenomonadaceae*, is predominantly found in the rumen, human oral cavity, and cecum of mammals. The oral *Selenomonas* species, particularly *Selenomonas sputigena*, can adhere to gingival epithelial cells and induce the secretion of MMPs and pro-inflammatory cytokines (45). Through microbial interactions and subgingival biofilm formation (46, 47), it actively contributes to the development of periodontitis (48, 49). *Capnocytophaga*, considered a periodontal pathogen (50), is an early colonizer of dental plaque. Studies suggest that species proportion changes within this genus can serve as dynamic markers of periodontitis progression, although it has also been detected in the periodontal health of young individuals (51, 52). *Capnocytophaga ochracea* is a significant member of the oral biofilm (53), with its bacterial outer membrane vesicles potentially playing a role in periodontitis and systemic disease development (54). Directly linking oral microbiota to BPH remains challenging. This study

revealed a significant enrichment of *Selenomonas* and *Capnocytophaga* in the oral cavity of BPH patients. These findings align with our previous research, where we concurrently detected *Selenomonadaceae* and *Capnocytophaga ochracea* in both subgingival plaque and prostatic fluid from BPH patients with periodontitis (18). These results suggest that an increased burden of specific oral bacteria may translocate to the prostate and contribute to the pathogenesis of BPH.

In patients diagnosed with both periodontitis and BPH, oral *Filifactor alocis* levels were significantly elevated compared to those with periodontitis alone. Additionally, a notable positive correlation was found between IPSS, PV, and *Filifactor alocis*, suggesting its potential role in the pathogenesis and progression of BPH. *Filifactor alocis* is frequently detected in patients with periodontitis (55). It promotes the colonization and invasion of other periodontal pathogens (56, 57) and can disseminate to organs such as the lungs, spleen, and kidneys, where it exerts pro-inflammatory and tissue-damaging effects (58). Persistent chronic inflammation and subsequent tissue damage repair are key factors in initiating and sustaining hyperplasia (59).

Earlier studies have shown that metabolites associated with periodontitis are mainly amino acids, organic acids, and their derivatives (60–62). Our findings align with previous research, showing that the primary differential metabolites in the periodontitis and P-BPH groups, compared with the Healthy group, include phosphatidylethanolamine (0:0/16:1(9Z)) and dipeptides, such as leucyl-leucine, histidyl-tyrosine, and phenylalanyl-phenylalanine. These metabolites showed a significant positive correlation with periodontal pathogens such as *Filifactor alocis*, *Treponema denticola*, and *Selenomonas dianae*. Arachidonic acid (AA) was significantly upregulated in the P-BPH group compared with the Healthy group, while adrenic acid was elevated in both the BPH and P-BPH groups. Adrenic acid, an extension product of arachidonic acid, plays a crucial role as a polyunsaturated fatty acid. AA metabolism via cyclooxygenase, lipoxygenase, and cytochrome P450 enzyme pathways contributes to inflammation, aging, and metabolic disorders. In BPH patients, serum AA metabolites were significantly elevated (63). Testosterone-induced BPH rats also showed increased prostate AA levels (64). AA metabolic inhibition attenuated BPH-associated inflammation (65).

This study represents a pioneering investigation in its field. However, it had some limitations. First, the samples were recruited from a single hospital, and the sample size was relatively small, and the results may be unstable with large uncertainty. We conducted post-hoc power analysis based on the current sample sizes and found that 16.22% of differentially abundant microbial features ($n = 74$) and 40.63% of differential metabolites ($n = 96$) achieved a power above 0.8. These results suggest that the current sample size is partially adequate, while larger sample sizes are still needed in future studies to achieve sufficient power (>0.8) for a higher proportion of microbial and metabolic features. Nevertheless, this research remains a valuable exploratory insight for sample size estimation of future studies. Future studies with large sample sizes and multicenter designs are warranted to increase statistical power and generalization. Second, the association between oral microbiome or metabolome profiles and BPH remains observational, and mechanistic studies are needed to explain how these compounds contribute to local or systemic inflammation, so as to establish a causality relationship. This study focused on a Chinese population. Variations in microbiota may exist among different races and dietary habits, which could potentially limit the generalizability of our findings. We recommend conducting similar studies on populations with different ethnic backgrounds and dietary patterns to further explore this topic.

## Conclusion

Significant differences exist in the periodontal microbiota composition and metabolites among BPH patients, BPH combined with periodontitis patients, periodontitis patients, and healthy individuals. BPH patients exhibit higher abundance of *Selenomonas* and *Capnocytophaga* in their oral cavities. The activation of arachidonic acid-related

inflammatory pathways may represent a key mechanism linking periodontitis and BPH, and future studies should focus on investigating related mechanisms.

## ACKNOWLEDGMENTS

This work was partially supported by the National Key Research and Development Program for Young Scientists, Grant/Award Number: 2022YFC3600700 (X.-T.Z.); Fundamental Research Funds for the Central Universities, Grant/Award Number: 2042025kf0095 (X.-T.Z.); National Natural Science Foundation of China, Grant/Award Number: 82200862 (C.F.); National Natural Science Foundation of China, Grant/Award Number: 82370778 (X.-T.Z.); The Young Top-notch Talent Cultivation Program of Hubei Province (X.-T.Z.); Hubei Provincial Natural Science Foundation of China, Grant/Award Number: 2023AFA061 (X.-T.Z.).

X.-T.Z. and C.F. conceived and designed the study. C.Z., C.L., L.W., Y.-Y.Z., Y.Y., and J.Y. collected clinical samples. C.F., L.-Y.L., S.-Y.W., and C.Z. performed data analysis. C.Z. wrote the manuscript. X.-T.Z., C.F., and L.-M.X. critically revised the manuscript. All authors reviewed the manuscript and gave final approval to be accountable for all aspects of the work.

## AUTHOR AFFILIATIONS

[1]Center for Evidence-Based and Translational Medicine, Zhongnan Hospital of Wuhan University, Wuhan, China

[2]Department of Urology, Department of Geriatrics, Hubei Key Laboratory of Urinary System Diseases, Zhongnan Hospital of Wuhan University, Wuhan, China

[3]Department of Gastrointestinal Surgery, The First Affiliated Hospital of Guilin Medical University, Guilin, Guangxi, China

[4]Department of Pediatric Surgery, The First Affiliated Hospital of Zhengzhou University, Zhengzhou, Henan, China

[5]Department of Stomatology, Zhongnan Hospital of Wuhan University, Wuhan, China

[6]Physical Examination Center, Integrated Chinese and Western Medicine Hospital of XiangYang (Dongfeng People's Hospital), Xiangyang, China

[7]Department of Urology, The First People's Hospital of Tianmen in Hubei Province, The Affiliated Hospital of Hubei University of Science and Technology, Tianmen, China

[8]Department of Epidemiology and Biostatistics, School of Public Health, Wuhan University, Wuhan, China

## AUTHOR ORCIDs

Cong Zhu http://orcid.org/0009-0009-1428-2823
Xian-Tao Zeng http://orcid.org/0000-0003-1262-725X
Cheng Fang http://orcid.org/0000-0002-8626-9451

## DATA AVAILABILITY

Sequencing data are openly available in the NCBI Sequence Read Archive (SRA) under BioProject accession number PRJNA1289417.

## ETHICS APPROVAL

All participants were provided with both written and oral information prior to the commencement of the study, and each signed an informed consent form. This study protocol was approved by the Ethics Committee of Zhongnan Hospital of Wuhan University (2022173), and the study was conducted in accordance with the Helsinki Declaration of 1975, as revised in 2013.

## ADDITIONAL FILES

The following material is available online.

### Supplemental Material

**Fig. S1 (Spectrum03376-24-S0001.tif).** Species composition of oral microbiota among the Healthy, BPH, Periodontitis, and P-BPH groups.
**Fig. S2 (Spectrum03376-24-S0002.tif).** LDA effect size (LEfSe) analysis was used to compare the different microorganisms among the four groups.
**Fig. S3 (Spectrum03376-24-S0003.tif).** Correlation between differential metabolite and clinical parameters.
**Supplemental material (Spectrum03376-24-S0004.docx).** Supplemental figure legends.
**Supplemental tables (Spectrum03376-24-S0005.docx).** Tables S1 to S3.

### Open Peer Review

**PEER REVIEW HISTORY (review-history.pdf).** An accounting of the reviewer comments and feedback.

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
