## [Reviewer comments · Microbiology Spectrum]

Microbiology Spectrum

Profiles of oral microbiota and metabolites in periodontitis and benign prostatic hyperplasia patients: a pilot study

Cong Zhu, Lu-Yao Li, Cheng Li, Lan Wu, Yuan-Yuan Zhang, Yan Yao, Jun Yang, Shuang-Ying Wang, Li-Min Xing, Xian-Tao Zeng, and Cheng Fang

Corresponding Author(s): Cheng Fang, Wuhan University

Review Timeline:

Submission Date:	December 23, 2024
Editorial Decision:	April 18, 2025
Revision Received:	June 9, 2025
Editorial Decision:	June 27, 2025
Revision Received:	July 23, 2025
Accepted:	July 24, 2025

Editor: Zhenjiang Xu

Reviewer(s): Disclosure of reviewer identity is with reference to reviewer comments included in decision letter(s). The following individuals involved in review of your submission have agreed to reveal their identity: Jôice Dias Corrêa (Reviewer #1); Deeksha Tripathi (Reviewer #2)

Transaction Report:

DOI: <https://doi.org/10.1128/spectrum.03376-24>

Re: Spectrum03376-24 (Profiles of oral microbiota and metabolites in periodontitis and benign prostatic hyperplasia patients: a pilot study)

Dear Dr. Cheng Fang:

Thank you for the privilege of reviewing your work. Below you will find my comments, instructions from the Spectrum editorial office, and the reviewer comments.

Revision Guidelines

Sincerely,
Zhenjiang Xu
Editor
Microbiology Spectrum

Reviewer #1 (Comments for the Author):

The periodontal examination was conducted on only one tooth per sextant. In microbiome collection cases, it is necessary to analyze all sites, as it is known that inflammatory changes in one tooth can affect the functional responses of the microbiome.

All participants were recruited from a single hospital (Xiangyang Integrated Traditional Chinese and Western Medicine Hospital).

This may limit the generalizability of the results to other populations or geographic contexts. It is recommended to discuss this limitation in the "Discussion" section of the article.

There is no mention of the recruitment method (e.g., random or convenience sampling), which is crucial for evaluating potential biases.

It is unclear whether the periodontal parameters were measured by a single calibrated examiner or by a team. This is essential to ensure consistency and reduce inter-examiner bias. It is suggested to describe the calibration process.

The phrase "functional teeth or those with more severe periodontal conditions" does not specify whether the collection was limited to a single tooth per patient or conducted on multiple teeth. It is recommended to clarify the total number of teeth and segments included.

The type of solution used for mouth rinsing (e.g., water or antiseptic) and the time between rinsing and sample collection are not detailed. These factors can influence the composition of gingival fluid, especially the microbial load.

The selection of "severest" teeth may introduce bias if the inflammation levels are not representative of the patient's entire periodontal environment. Alternatively, collecting samples from teeth with medium periodontal conditions (not just the extremes) could provide more generalizable data.

There is no mention of DNA quality control before PCR amplification. Methods such as spectrophotometry (e.g., 260/280 ratio) or gel electrophoresis could be included to confirm DNA purity and integrity.

It is unclear whether negative controls were used during DNA extraction and PCR amplification to detect contamination. These controls are essential in microbiological studies.

The choice to group readings into OTUs with 97% similarity, although standard, is gradually being replaced by methods based on amplicon sequence variants (ASVs), which are more robust and offer greater taxonomic resolution. Considering the use of tools like DADA2 or Deblur would be more modern and reliable.

The text does not specify the statistical significance threshold for the LEfSe test. It is recommended to report the values used (e.g., LDA threshold > 2.0) to ensure transparency.

While robust databases were mentioned, the criteria for metabolite identification (e.g., m/z, RT, or MS/MS spectra) were not described. Specifying whether a spectral similarity threshold was used, for example, would increase clarity.

The exclusion of metabolites with CV > 30% is valid but may lead to the loss of relevant biological information, especially in exploratory studies. Considering presenting data on the proportion of metabolites excluded would be helpful. I suggest reporting the proportion of metabolites excluded due to high CV and justifying how this exclusion did not compromise the biological interpretations.

It is recommended to discuss how *Porphyromonas gingivalis* and *Treponema medium* could contribute to the inflammation associated with BPH, based on previous studies.

Emphasize that the reported associations are observational and that further studies are needed to confirm a causal relationship.

The association between metabolites and microorganisms is interesting, but mechanistic analyses are missing to explain how these compounds contribute to local or systemic inflammation.

While the role of *Selenomonas sputigena* and *Capnocytophaga ochracea* in periodontitis is well described, their specific relationship with BPH is only suggested, without detailed explanation or experimental basis.

The small sample size is a critical limitation that undermines the robustness of the findings. The text acknowledges these important limitations, such as the small sample size and the restriction of the study to a Chinese population, highlighting the need for validation in other populations. However, details about statistical power calculations could be presented to assess whether the observed correlations are reliable.

Reviewer #2 (Comments for the Author):

Major Review Questions

1. Sample Size Justification

- The study uses 12 subjects per group (n=48 total). What power analysis was conducted to determine this sample size? Small cohorts risk underpowered conclusions, especially given the multifactorial nature of BPH and periodontitis.

2. Confounding Variables

- Table 1 shows a near-significant age difference ($p=0.078$) between the P-BPH group (mean age 50.08) and healthy controls (46.25). How was age controlled in analyses? Older age is a known risk factor for both BPH and periodontitis, potentially biasing results.

3. Causality vs. Association

- The study identifies microbial/metabolite differences but uses a cross-sectional design. How do the authors plan to address the limitation that correlation \neq causation in future work?

4. Clinical Relevance of Metabolites

- Arachidonic acid and adrenic acid are highlighted as upregulated in BPH/P-BPH groups. Do these metabolites have established mechanistic links to prostate hyperplasia, or is this exploratory?

5. Generalizability

- Participants were recruited from a single hospital in China. How might regional, genetic, or lifestyle factors (e.g., diet, oral hygiene practices) limit the global applicability of findings?

Minor Review Questions

1. Abbreviations Clarity

- "IIEF" and "FPSA" are used in Table 1 but not defined. Please expand these terms (International Index of Erectile Function; Free Prostate-Specific Antigen).

2. Grammar/Syntax

- Abstract Line 43: "need more active periodontal treatment" \rightarrow "may benefit from more active periodontal treatment" to avoid overstatement.

- Importance Section Line 48: "unclearly" \rightarrow "unclear" ("The pathogenesis of BPH remains unclear").

3. Data Presentation

- Table 1: The "Teeth number" row for the P-BPH group shows "0.30" without a standard deviation (\pm). Is this an error?

4. Methodological Detail

- Line 126-127: Specify the volume/type of cryotubes used for GCF storage and whether protease inhibitors were added to prevent metabolite degradation.

5. Figure/Table Referencing

- Results mention "Table S1" (line 184) and "Figure 5," but supplemental materials are not included in the provided text. Ensure all referenced data is accessible to reviewers.

Major Review Questions

1. Sample Size Justification

- The study uses 12 subjects per group (n=48 total). What power analysis was conducted to determine this sample size? Small cohorts risk underpowered conclusions, especially given the multifactorial nature of BPH and periodontitis.

2. Confounding Variables

- Table 1 shows a near-significant age difference (p=0.078) between the P-BPH group (mean age 50.08) and healthy controls (46.25). How was age controlled in analyses? Older age is a known risk factor for both BPH and periodontitis, potentially biasing results.

3. Causality vs. Association

- The study identifies microbial/metabolite differences but uses a cross-sectional design. How do the authors plan to address the limitation that correlation \neq causation in future work?

4. Clinical Relevance of Metabolites

- Arachidonic acid and adrenic acid are highlighted as upregulated in BPH/P-BPH groups. Do these metabolites have established mechanistic links to prostate hyperplasia, or is this exploratory?

5. Generalizability

- Participants were recruited from a single hospital in China. How might regional, genetic, or lifestyle factors (e.g., diet, oral hygiene practices) limit the global applicability of findings?

Minor Review Questions

1. Abbreviations Clarity

- "IIEF" and "FPSA" are used in Table 1 but not defined. Please expand these terms (International Index of Erectile Function; Free Prostate-Specific Antigen).

2. Grammar/Syntax

- Abstract Line 43: "need more active periodontal treatment" → "may benefit from more active periodontal treatment" to avoid overstatement.

- Importance Section Line 48: "unclearly" → "unclear" ("The pathogenesis of BPH remains unclear").

3. Data Presentation

- Table 1: The "Teeth number" row for the P-BPH group shows "0.30" without a standard deviation (\pm). Is this an error?

4. Methodological Detail

- Line 126-127: Specify the volume/type of cryotubes used for GCF storage and whether protease inhibitors were added to prevent metabolite degradation.

5. Figure/Table Referencing

- Results mention "Table S1" (line 184) and "Figure 5," but supplemental materials are not included in the provided text. Ensure all referenced data is accessible to reviewers.

Overall Impression: The study addresses an innovative oral-prostate axis hypothesis with rigorous multi-omics methods. Addressing these questions would strengthen its contribution to understanding BPH pathophysiology and periodontal-systemic connections.

Response to Reviewers' Comments

Dear Editors,

Thank you for processing our manuscript entitled “Profiles of oral microbiota and metabolites in periodontitis and benign prostatic hyperplasia patients: a pilot study” (Manuscript ID Spectrum03376-24). We have completed the revisions and submitted our revised manuscript: one with changes highlighted and another one without any marks. We have thoroughly revised our manuscript by addressing all the points raised by the reviewers, and have highlighted all modifications in red font. Along with the revised manuscript, we also submit a response letter outlining our responses to the reviewers' comments and the changes we have made. Please find below our point-by-point responses to the reviewers' comments.

Reviewer 1:

Comment 1. The periodontal examination was conducted on only one tooth per sextant. In microbiome collection cases, it is necessary to analyze all sites, as it is known that inflammatory changes in one tooth can affect the functional responses of the microbiome.

Response :

Thank you for your comments and valuable suggestions. The partial-mouth periodontal examination (one tooth per sextant) is a simple and highly repeatable method for evaluating periodontal health status. It can well represent the periodontal state of the entire mouth and has been widely applied to studies. The references are as follows:

[1] Jensen ED, Selway CA, Allen G, Bednarz J, Weyrich LS, Gue S, Peña AS, Couper J. Early markers of periodontal disease and altered oral microbiota are associated with glycemic control in children with type 1 diabetes. *Pediatr Diabetes*. 2021 May;22(3):474-481. doi: 10.1111/pedi.13170. Epub 2020 Dec 15. PMID: 33398933.

[2] Selway CA, Jensen ED, Pena AS, Smart G, Weyrich LS. Type 1 diabetes, periodontal health, and a familial history of hyperlipidaemia is associated with oral microbiota in children: a cross-sectional study. *BMC Oral Health*. 2023 Jan 11;23(1):15. doi: 10.1186/s12903-022-02625-0. PMID: 36631887; PMCID: PMC9832783.

[3] Duan X, Chen X, Gupta M, Seriwatanachai D, Xue H, Xiong Q, Xu T, Li D, Mo A, Tang X, Zhou X, Li Y, Yuan Q. Salivary microbiome in patients undergoing hemodialysis and its associations with the duration of the dialysis. *BMC Nephrol*. 2020 Sep 29;21(1):414. doi: 10.1186/s12882-020-02009-y. PMID: 32993533; PMCID: PMC7523083.

Relative revision in manuscript:

(Methods) Page 3-4, line 109-113. "The oral cavity was divided into six segments, with at least one functional tooth selected from each segment, excluding the third molars. If the chosen functional tooth in a segment was missing, it was replaced by a neighboring tooth; if the entire segment lacked functional teeth, it was omitted. Each patient was detected 6 segments, a total of 6 teeth."

Comment 2. All participants were recruited from a single hospital (Xiangyang Integrated Traditional Chinese and Western Medicine Hospital). This may limit the generalizability of the results to other populations or geographic contexts. It is recommended to discuss this limitation in the "Discussion" section of the article.

Response :

Thank you for your comments and valuable suggestions. We conducted post-hoc power analysis based on the current sample sizes and found that 16.22% of differentially abundant microbial features (n=74) and 40.63% of differential metabolites (n=96) achieved a power above 0.8. These results suggest that the current sample size is partially adequate, while larger sample sizes are still needed in future studies to achieve sufficient power (>0.8) for a higher proportion of microbial and

metabolic features, and the results may be unstable with large uncertainty. We have addressed the sample limitations in the Discussion section.

Relative revision in manuscript:

(Discussion) Page 12, line 452-463. “However, it had some limitations. First, the samples were recruited from a single hospital, and the sample size was relatively small, and the results may be unstable with large uncertainty. We conducted post-hoc power analysis based on the current sample sizes and found that 16.22% of differentially abundant microbial features (n=74) and 40.63% of differential metabolites (n=96) achieved a power above 0.8. These results suggest that the current sample size is partially adequate, while larger sample sizes are still needed in future studies to achieve sufficient power (>0.8) for a higher proportion of microbial and metabolic features. Nevertheless, this research remains a valuable exploratory insight for sample size estimation of future studies. Future study with large sample size and multicenter are warranted to increase statistical power and generalization.”

Comment 3. There is no mention of the recruitment method (e.g., random or convenience sampling), which is crucial for evaluating potential biases.

Response :

Thank you for your comments and valuable suggestions. We have made corresponding additions to the Methods section.

Relative revision in manuscript:

(Methods) Page 3, line 94-95. “Participants were consecutively enrolled from Xiangyang Integrated Traditional Chinese and Western Medicine Hospital during June to October 2023.”

Comment 4. It is unclear whether the periodontal parameters were measured by a single calibrated examiner or by a team. This is essential to ensure consistency and reduce inter-examiner bias. It is suggested to describe the calibration process.

Response :

Thank you for your comments and valuable suggestions. We have enhanced the Methods section with more comprehensive explanations of our oral parameter measurement protocols and periodontal assessment methodology.

Relative revision in manuscript:

(Methods) Page 4, line 124-127. "All oral parameter assessments were performed by two qualified dental specialists who independently diagnosed and verified the final oral health status of each participant, after a consensus check prior to sample collection to ensure consistency of findings."

Comment 5. The phrase "functional teeth or those with more severe periodontal conditions" does not specify whether the collection was limited to a single tooth per patient or conducted on multiple teeth. It is recommended to clarify the total number of teeth and segments included.

Response :

Thank you for your comments and valuable suggestions. We have added a more detailed description in the Methods section.

Relative revision in manuscript:

(Methods) Page 3-4, line 109-113. "The oral cavity was divided into six segments, with at least one functional tooth selected from each segment, excluding the third molars. If the chosen functional tooth in a segment was missing, it was replaced by a neighboring tooth; if the entire segment lacked functional teeth, it was omitted. Each patient was detected 6 segments, a total of 6 teeth."

Comment 6. The type of solution used for mouth rinsing (e.g., water or antiseptic) and the time between rinsing and sample collection are not detailed. These factors can influence the composition of gingival fluid, especially the microbial load.

Response :

Thank you for your comments and valuable suggestions. We have added a more detailed description in the Methods section.

Relative revision in manuscript:

(Methods) Page 4, line 128-131. "After rinsing with sterile saline, the sampling area was isolated with cotton rolls and dried. Gingival crevicular fluid (GCF) samples were immediately collected by the dentist from four sites on six functional teeth, or the alternative teeth with the worse periodontal condition when the specified functional teeth loss."

Comment 7. The selection of "severest" teeth may introduce bias if the inflammation levels are not representative of the patient's entire periodontal environment. Alternatively, collecting samples from teeth with medium periodontal conditions (not just the extremes) could provide more generalizable data.

Response :

Thank you for your comments and valuable suggestions. As described in the manuscript, the definition of periodontitis is not that all teeth suffered periodontal inflammation, but some teeth have periodontal problems. The periodontal microbiota of the periodontitis should be analyzed from the teeth with periodontitis. Thus, we want to emphasize that when the specified functional teeth loss, among the adjacent candidate teeth in the area, the tooth with a poorer periodontal condition will be selected, which can better reflect the periodontal microbiota status of periodontal disease. The expression in the text is incorrect and relevant corrections have been made. We have added a more detailed description in the Methods section.

Relative revision in manuscript:

(Methods) Page 4, line 128-131. "After rinsing with sterile saline, the sampling area was isolated with cotton rolls and dried. Gingival crevicular fluid (GCF) samples were immediately collected by the dentist from four sites on six functional teeth, or the alternative teeth with the worse periodontal condition when the specified functional teeth loss."

Comment 8. There is no mention of DNA quality control before PCR amplification.

Methods such as spectrophotometry (e.g., 260/280 ratio) or gel electrophoresis could be included to confirm DNA purity and integrity.

Response :

Thank you for your comments and valuable suggestions. We have added DNA quality control instructions in the Methods section.

Relative revision in manuscript:

(Methods) Page 4, line 140-144. “DNA quality control was performed using a fluorescent dye-based method. This approach quantifies DNA concentration by measuring the fluorescence intensity generated when the dye binds to DNA. The method features simple operation and reliable results, thereby effectively guaranteeing the quality of DNA templates for subsequent PCR amplification.”

Comment 9. It is unclear whether negative controls were used during DNA extraction and PCR amplification to detect contamination. These controls are essential in microbiological studies.

Response :

Thank you for your comments and valuable suggestions. We have added DNA quality control instructions in the Methods section

Relative revision in manuscript:

(Methods) Page 4, line 146-149. “During DNA extraction and PCR amplification procedures, we strictly adhered to standardized experimental protocols. Sterile nuclease-free water was systematically included as blank controls to monitor potential contamination throughout the experimental process.”

Comment 10. The choice to group readings into OTUs with 97% similarity, although standard, is gradually being replaced by methods based on amplicon sequence variants (ASVs), which are more robust and offer greater taxonomic resolution. Considering the use of tools like DADA2 or Deblur would be more modern and reliable.

Response :

We sincerely appreciate your valuable and constructive suggestion regarding the use of DADA2 for ASVs analysis. We carefully tested this method on our raw sequencing data, and while we recognize its advantages, we ultimately retained the OTUs clustering approach for this study. This decision was based on our extensive experience with OTU-based pipelines, which ensures robust and efficient data processing, as well as the need to meet the timeline for this publication.

We fully agree that ASV methods represent an important advancement in microbiome research, and we are committed to incorporating them in future studies as we further refine our expertise. Thank you again for your insightful comment, it has encouraged us to prioritize methodological updates moving forward.

Comment 11. The text does not specify the statistical significance threshold for the LEfSe test. It is recommended to report the values used (e.g., LDA threshold > 2.0) to ensure transparency.

Response :

Thank you for your comments and valuable suggestions. We have added the statistical significance threshold for the LEfSe test in the Methods section.

Relative revision in manuscript:

(Methods) Page 5, line 165-168. “Inter-group microbial comparisons were performed using linear discriminant analysis (LDA) effect size (LEfSe) analysis implemented in the microeco package (v1.9.1), with significance thresholds set at LDA score > 2 and p-value < 0.05”.

Comment 12. While robust databases were mentioned, the criteria for metabolite identification (e.g., m/z, RT, or MS/MS spectra) were not described. Specifying whether a spectral similarity threshold was used, for example, would increase clarity.

Response :

Thank you for your comments and valuable suggestions. We have added a more detailed description in the Methods section.

Relative revision in manuscript:

(Methods) Page 5, line 175-181. “Mass spectrometry data were imported into Compound Discoverer 3.3 software and analyzed with the BMDB (BGI Metabolome Database), mzCloud, and ChemSpider databases to generate a data matrix containing metabolite peak areas and identification results (Parameters: Parent ion mass deviation: <5ppm, Fragment ion mass deviation: <10ppm Retention time deviation: <0.2min). Then, the secondary mass spectrum score was used to classify the level and make a credibility assessment of the identified metabolites.”

Comment 13. The exclusion of metabolites with CV > 30% is valid but may lead to the loss of relevant biological information, especially in exploratory studies. Considering presenting data on the proportion of metabolites excluded would be helpful. I suggest reporting the proportion of metabolites excluded due to high CV and justifying how this exclusion did not compromise the biological interpretations.

Response :

Thank you for your comments and valuable suggestions. We have added a more detailed description in the Results section.

Relative revision in manuscript:

(Results) Page 9, line 295-299. “A total of 547 metabolites (8%) were ultimately excluded. These excluded high-CV features primarily representing low-abundance or noise-prone compounds with unstable quantification. In contrast, Retaining $CV \leq 30\%$ features significantly improved statistical power and reproducibility, ensuring reliable differential metabolite identification and pathway analysis.”

Comment 14. It is recommended to discuss how *Porphyromonas gingivalis* and *Treponema medium* could contribute to the inflammation associated with BPH, based on previous studies.

Response :

Thank you for your comments and valuable suggestions. We have added the potential relationship between *Porphyromonas gingivalis*, *Treponema pallidum*, and the pathogenesis of BPH in the revised Discussion section.

Relative revision in manuscript:

(Discussion) Page 11, line 383-388. “Our published findings demonstrated that *Porphyromonas gingivalis* infection significantly upregulates IL-6 expression in prostate tissue and activates the PI3K/AKT signaling pathway, thereby modulating prostate cell proliferation and apoptosis¹⁸. *Treponema* medium, a Gram-negative anaerobic spirochete, has been detected in dental plaque from periodontitis patients. However, direct evidence linking this oral pathogen to BPH pathogenesis remains lacking²⁷.”

Comment 15. Emphasize that the reported associations are observational and that further studies are needed to confirm a causal relationship.

Response :

Thank you for your comments and valuable suggestions. We have modified the discussion section accordingly.

Relative revision in manuscript:

(Discussion) Page 13, line 463-466. “Second, the association between oral microbiome or metabolome profiles and BPH remains observational, and mechanistic studies are needed to explain how these compounds contribute to local or systemic inflammation, so as to establish a causality relationship.”

Comment 16. The association between metabolites and microorganisms is interesting, but mechanistic analyses are missing to explain how these compounds contribute to local or systemic inflammation.

Response :

Thank you for your comments and valuable suggestions. We have modified the discussion section accordingly. Building upon our laboratory's established research framework investigating the relationship between periodontitis and BPH, we have developed preliminary methodologies and obtained supporting evidence. As demonstrated in our published study (PMID: 38764065), we have employed *Porphyromonas gingivalis* (*P. gingivalis*) lipopolysaccharide to stimulate prostate cells, establishing various experimental models including rat periodontitis models, BPH models, and *P. gingivalis*-induced BPH models to examine their causal relationship. In the current study, we have identified specific oral microbiota and metabolites associated with BPH. These findings can be further validated in future research using similar experimental approaches.

Relative revision in manuscript:

(Discussion) Page 13, line 463-466. "Second, the association between oral microbiome or metabolome profiles and BPH remains observational, and mechanistic studies are needed to explain how these compounds contribute to local or systemic inflammation, so as to establish a causality relationship."

Comment 17. While the role of selenomonadaceae *Selenomonas sputigena* and *Capnocytophaga ochracea* in periodontitis is well described, their specific relationship with BPH is only suggested, without detailed explanation or experimental basis.

Response :

Thank you for your comments and valuable suggestions. We have revised and supplemented the Discussion section.

Relative revision in manuscript:

(Discussion) Page 11, line 406-413. "In patients with prostatic hyperplasia, we observed a significant increase in *Selenomonas* and *Capnocytophaga* in their oral cavity compared to the normal population. *Selenomonas*, a genus within the family *Selenomonadaceae*, is predominantly found in the rumen, human oral cavity, and cecum of mammals. The oral *Selenomonas* species, particularly *Selenomonas*

sputigena, can adhere to gingival epithelial cells and induce the secretion of MMPs and pro-inflammatory cytokines⁴⁵. Through microbial interactions and subgingival biofilm formation^{46,47}, it actively contributes to the development of periodontitis^{48,49}.”

(Discussion) Page 12, line 419-426. “Directly linking oral microbiota to BPH remains challenging. This study revealed a significant enrichment of *Selenomonas* and *Capnocytophaga* in the oral cavity of BPH patients. These findings align with our previous research, where we concurrently detected *Selenomonadaceae* and *Capnocytophaga ochracea* in both subgingival plaque and prostatic fluid from BPH patients with periodontitis¹⁸. These results suggest that an increased burden of specific oral bacteria may translocate to the prostate and contribute to the pathogenesis of BPH.”

Comment 18. The small sample size is a critical limitation that undermines the robustness of the findings. The text acknowledges these important limitations, such as the small sample size and the restriction of the study to a Chinese population, highlighting the need for validation in other populations. However, details about statistical power calculations could be presented to assess whether the observed correlations are reliable.

Response :

Thank you for your comments and valuable suggestions. We conducted post-hoc power analysis based on the current sample sizes and found that 16.22% of differentially abundant microbial features (n=74) and 40.63% of differential metabolites (n=96) achieved a power above 0.8. These results suggest that the current sample size is partially adequate, while larger sample sizes are still needed in future studies to achieve sufficient power (>0.8) for a higher proportion of microbial and metabolic features, and the results may be unstable with large uncertainty. We have addressed the sample limitations in the Discussion section.

Relative revision in manuscript:

(Discussion) Page 12, line 452-463. “However, it had some limitations. First, the

samples were recruited from a single hospital, and the sample size was relatively small, and the results may be unstable with large uncertainty. We conducted post-hoc power analysis based on the current sample sizes and found that 16.22% of differentially abundant microbial features (n=74) and 40.63% of differential metabolites (n=96) achieved a power above 0.8. These results suggest that the current sample size is partially adequate, while larger sample sizes are still needed in future studies to achieve sufficient power (>0.8) for a higher proportion of microbial and metabolic features. Nevertheless, this research remains a valuable exploratory insight for sample size estimation of future studies. Future study with large sample size and multicenter are warranted to increase statistical power and generalization.”

Reviewer 2:

Comment 1. Sample Size Justification- The study uses 12 subjects per group (n=48 total). What power analysis was conducted to determine this sample size? Small cohorts risk underpowered conclusions, especially given the multifactorial nature of BPH and periodontitis.

Response :

Thank you for your comments and valuable suggestions. We conducted post-hoc power analysis based on the current sample sizes and found that 16.22% of differentially abundant microbial features (n=74) and 40.63% of differential metabolites (n=96) achieved a power above 0.8. These results suggest that the current sample size is partially adequate, while larger sample sizes are still needed in future studies to achieve sufficient power (>0.8) for a higher proportion of microbial and metabolic features, and the results may be unstable with large uncertainty. We have addressed the sample limitations in the Discussion section.

Relative revision in manuscript:

(Discussion) Page 12, line 452-463. “However, it had some limitations. First, the samples were recruited from a single hospital, and the sample size was relatively

small, and the results may be unstable with large uncertainty. We conducted post-hoc power analysis based on the current sample sizes and found that 16.22% of differentially abundant microbial features (n=74) and 40.63% of differential metabolites (n=96) achieved a power above 0.8. These results suggest that the current sample size is partially adequate, while larger sample sizes are still needed in future studies to achieve sufficient power (>0.8) for a higher proportion of microbial and metabolic features. Nevertheless, this research remains a valuable exploratory insight for sample size estimation of future studies. Future study with large sample size and multicenter are warranted to increase statistical power and generalization.”

Comment 2. Confounding Variables- Table 1 shows a near-significant age difference (p=0.078) between the P-BPH group (mean age 50.08) and healthy controls (46.25). How was age controlled in analyses? Older age is a known risk factor for both BPH and periodontitis, potentially biasing results.

Response :

Thank you for your comments and valuable suggestions. The P-BPH and healthy control groups showed borderline significant differences in age, with a mean intergroup difference of less than 4 years. This minor age disparity is clinically acceptable, since 4 years represents only a small fraction of the progression period for both periodontal disease and BPH. Nevertheless, we anticipate that future large-sample studies will help address potential age-related biases.

Relative revision in manuscript:

(Results) Page 6, line 199-203. “Demographic characteristics and clinical indicators indicate that there were no significant differences among the four groups in terms of age, BMI, education level, smoking, alcohol consumption, tea consumption, International index of erectile function (IIEF), prostate specific antigen (PSA) and free prostate specific antigen (FPSA)”.

Comment 3. Causality vs. Association- The study identifies microbial/metabolite

differences but uses a cross-sectional design. How do the authors plan to address the limitation that correlation \neq causation in future work?

Response :

Thank you for your comments and valuable suggestions. We have modified the discussion section accordingly. Building upon our laboratory's established research framework investigating the relationship between periodontitis and BPH, we have developed preliminary methodologies and obtained supporting evidence. As demonstrated in our published study (PMID: 38764065), we have employed *Porphyromonas gingivalis* (*P. gingivalis*) lipopolysaccharide to stimulate prostate cells, establishing various experimental models including rat periodontitis models, BPH models, and *P. gingivalis*-induced BPH models to examine their causal relationship. In the current study, we have identified specific oral microbiota and metabolites associated with BPH. These findings can be further validated in future research using similar experimental approaches.

Relative revision in manuscript:

(Discussion) Page 13, line 463-466. "Second, the association between oral microbiome or metabolome profiles and BPH remains observational, and mechanistic studies are needed to explain how these compounds contribute to local or systemic inflammation, so as to establish a causality relationship."

Comment 4. Clinical Relevance of Metabolites- Arachidonic acid and adrenic acid are highlighted as upregulated in BPH/P-BPH groups. Do these metabolites have established mechanistic links to prostate hyperplasia, or is this exploratory?

Response :

Thank you for your comments and valuable suggestions. We have addressed the sample limitations in the Discussion section.

Relative revision in manuscript:

(Discussion) Page 12, line 443-451. "Arachidonic acid (AA) was significantly upregulated in the P-BPH group compared to the healthy group, while adrenic acid

was elevated in both the BPH and P-BPH groups. Adrenic acid, an extension product of arachidonic acid, plays a crucial role as a polyunsaturated fatty acid. AA metabolism via cyclooxygenase, lipoxygenase, and cytochrome P450 enzyme pathways contributes to inflammation, aging, and metabolic disorders. In BPH patients, serum AA metabolites were significantly elevated⁶². Testosterone-induced BPH rats also showed increased prostate AA levels⁶³. AA metabolic inhibition attenuated BPH-associated inflammation⁶⁴.

Comment 5. Generalizability- Participants were recruited from a single hospital in China. How might regional, genetic, or lifestyle factors (e.g., diet, oral hygiene practices) limit the global applicability of findings?

Response :

Thank you for your comments and valuable suggestions. This study represents part of the data from our ongoing multicenter, large-sample clinical research (2022YFC3600700). Due to considerations such as research output translation and graduate student graduation timelines, we have opted to publish these preliminary findings in a timely manner. We have addressed the sample limitations in the Discussion section.

Relative revision in manuscript:

(Discussion) Page 12, line 452-454. "However, it had some limitations. First, the samples were recruited from a single hospital, and the sample size was relatively small, and the results may be unstable with large uncertainty."

Reviewer 2:

Comment 6. Abbreviations Clarity- "IIEF" and "FPSA" are used in Table 1 but not defined. Please expand these terms (International Index of Erectile Function; Free Prostate-Specific Antigen).

Response :

Thank you for your comments and valuable suggestions. We have modified the results section accordingly.

Relative revision in manuscript:

(Results) Page 6, line 199-203. “Demographic characteristics and clinical indicators indicate that there were no significant differences among the four groups in terms of age, BMI, education level, smoking, alcohol consumption, tea consumption, International index of erectile function (IIEF), prostate specific antigen (PSA) and free prostate specific antigen (FPSA)”

Comment 7. Grammar/Syntax- Abstract Line 43: "need more active periodontal treatment" → "may benefit from more active periodontal treatment" to avoid overstatement.

- Importance Section Line 48: "unclearly" → "unclear" ("The pathogenesis of BPH remains unclear").

Response :

Thank you for your comments and valuable suggestions. We have modified the Abstract and Importance section accordingly.

Relative revision in manuscript:

(Abstract) Page 2, line 49. “...especially those with periodontitis, may benefit from more active periodontal treatment.”

(Importance) Page 2, line 52. “The pathogenesis of BPH remains unclearly, with...”

Comment 8. Data Presentation- Table 1: The "Teeth number" row for the P-BPH group shows "0.30" without a standard deviation (plus minus). Is this an error?

Response :

Thank you for your comments and valuable suggestions. We have carefully checked and corrected the data in Table 1 in the article. See Table 1 for details.

Comment 9. Methodological Detail- Line 126-127: Specify the volume/type of cryotubes used for GCF storage and whether protease inhibitors were added to prevent metabolite degradation.

Response : Thank you for your comments and valuable suggestions. We have modified the results section accordingly.

Relative revision in manuscript:

(Methods) Page 4, line 135. “The absorbent paper points were placed into 2mL sterile enzyme-free cryotubes, sealed, and stored at -80°C.”

Comment 10. Figure/Table Referencing- Results mention "Table S1" (line 184) and "Figure 5," but supplemental materials are not included in the provided text. Ensure all referenced data is accessible to reviewers.

Response : Thank you for your comments and valuable suggestions. We have included all images and verified supplemental tables in our new submission.

Re: Spectrum03376-24R1 (Profiles of oral microbiota and metabolites in periodontitis and benign prostatic hyperplasia patients: a pilot study)

Dear Dr. Cheng Fang:

Thank you for the privilege of reviewing your work. Your manuscript is accepted in principle under the condition that the sequencing data and sample metadata are made publicly available, following ASM's open data policy (<https://journals.asm.org/open-data-policy>). Below you will find my comments, instructions from the Spectrum editorial office, and the reviewer comments.

Revision Guidelines

Sincerely,
Zhenjiang Xu
Editor
Microbiology Spectrum

Reviewer #1 (Comments for the Author):

Although the authors mention that the data are available upon request, it would be preferable to deposit the sequencing data in a public repository to ensure transparency and reproducibility.

All my other concerns have been addressed

Reviewer #2 (Comments for the Author):

The authors have addressed the reviewer comments thoughtfully and made appropriate revisions to the manuscript. The responses are clear, and the updated content strengthens the overall quality of the work. No further changes are recommended at this time. I appreciate the effort taken to improve the clarity and rigor of the submission.

Response to Reviewers' Comments

Dear Editors,

Thank you for processing our manuscript entitled “Profiles of oral microbiota and metabolites in periodontitis and benign prostatic hyperplasia patients: a pilot study” (Manuscript ID Spectrum03376-24R1). It is a great honor for us that the article has been accepted. Following ASM's open data policy, sequencing data in our study are now openly available in the NCBI Sequence Read Archive (SRA) under BioProject accession number PRJNA1289417. We have completed the revisions and submitted our revised manuscript: one with changes highlighted in red font and another one without any marks. Meanwhile, we have removed our supplemental legends for Supplemental figures from manuscript text file and uploaded into the system as a separate supplemental file.

Re: Spectrum03376-24R2 (Profiles of oral microbiota and metabolites in periodontitis and benign prostatic hyperplasia patients: a pilot study)

Dear Dr. Cheng Fang:

Your manuscript has been accepted, and I am forwarding it to the ASM production staff for publication. Your paper will first be checked to make sure all elements meet the technical requirements. ASM staff will contact you if anything needs to be revised before copyediting and production can begin. Otherwise, you will be notified when your proofs are ready to be viewed.

Sincerely,
Zhenjiang Xu
Editor
Microbiology Spectrum